# Dietary diversity, nutritional status and associated factors among lactating mothers visiting government health facilities at Dessie town, Amhara region, Ethiopia

**Awel Seid, Hirut Assaye Cherie**📛 *

Department of Applied Human Nutrition, Bahir Dar Institute of Technology, Bahir Dar University, Bahir Dar, Ethiopia

* hirutas2000@gmail.com

**Data Availability Statement:** All relevant data are within the paper and its Supporting information files.

## Abstract

### Background

Maternal undernutrition is one of the most common causes of maternal morbidity and mortality in developing countries. Severe undernutrition among mothers leads to reduced lactation performance which further contributes to an increased risk of infant mortality. However, data regarding nutritional status of lactating mothers at Dessie town and its surrounding areas is lacking. This study assessed dietary diversity, nutritional status and associated factors of lactating mothers visiting health facilities at Dessie town, Amhara region, Ethiopia.

### Methods

Institutional based cross-sectional study was conducted from March to April, 2017 among 408 lactating mothers. Systematic random sampling technique was employed to select the study participants. Data on socio-demographic and economic characteristics, health related characteristics, dietary diversity and food security status of participants were collected using interviewer administered questionnaire. Data were entered into EPI-INFO and analyzed using SPSS Version 22. Bivariate and multivariate analyses were performed to identify factors associated with dietary diversity and nutritional status of lactating mothers.

### Results

More than half (55.6%) of lactating mothers had inadequate dietary diversity (DDS<5.3) and about 21% were undernourished (BMI<18.5 kg/m$^2$). Household monthly income [AOR = 2.0, 95% CI (1.15, 3.65)], type of house [AOR = 1.8, 95% CI (1.15, 2.94)], nutrition information [AOR = 1.6, 95% CI (1.05, 2.61)] and household food insecurity [AOR = 1.8, 95% CI (1.05, 3.06)] were factors associated with dietary diversity of lactating mothers. Being young in age 15–19 years [AOR = 10.3, 95% CI (2.89, 36.39)] & 20–29 years [AOR = 3.4, 95% CI (1.57, 7.36)], being divorced/separated [AOR = 10.1, 95% CI (1.42, 72.06)], inadequate dietary diversity [AOR = 3.8, 95% CI (2.08, 7.03)] and household food insecurity [AOR = 3.1, 95% CI (1.81, 5.32)] were factors associated with maternal undernutrition.

**Funding:** The funding was in the form of financial support to help the researchers for their data collection. However, the funder had no role in study design, data collection and analysis, decision to publish, or preparation of the manuscript. Besides, any of the authors did not receive salary from the funder for this specific study.

**Competing interests:** The authors have declared that no competing interests exist.

**Abbreviations:** AOR, Adjusted odds ratio; BMI, Body mass Index; CI, Confidence interval; COR, Crude odds ratio; CSA, Central Statistics Agency; DDS, Dietary Diversity Score; ETB, Ethiopian Birr; WHO, World Health Organization.

## Conclusion

The dietary diversity of lactating mothers in the study area was sub optimal and the prevalence of undernutrition was relatively high. Public health nutrition interventions such as improving accessibility of affordable and diversified nutrient rich foods are important to improve the nutritional status of mothers and their children in the study area.

## Background

Nutrient requirements increase considerably during lactation since breast milk has to supply an adequate amount of all the nutrients for an infant's needs for growth and development [1]. Lactating women require approximately 500 additional kcal per day beyond what is recommended for non-pregnant women [2]. It is therefore important that lactating women eat sufficient quantity and quality of food during this period [3]. Nutritional inadequacy of lactating mothers not only affects milk composition and production but also the health of the mothers and their infants. If the mother is undernourished during lactation, the nutrients that are transferred to the baby will be of poor quality and quantity [4]. One of the proxy indicators for measuring dietary adequacy of lactating mothers is dietary diversity which refers to the number of different foods or food groups consumed over a given reference period [5].

According to the 2016 demographic and health survey, maternal mortality in Ethiopia is 412/100,000 live births [6]. Women of reproductive age are also vulnerable to undernutrition. The 2011 Ethiopian demographic and health survey revealed that the level of undernutrition among women is relatively high with 27% of women either thin or undernourished [7]. Studies conducted on lactating mothers from 2011 up to 2016 in different parts of Ethiopia also indicated their poor nutritional status [8–12] and poor dietary diversity [13, 14].

A number of factors were reported to be associated with mothers' dietary diversity; maternal education [13], monthly income, home gardening, source of drinking water [14], food security, maternal health [13, 15] and season [15]. Factors such as size of farm land, length of years of marriage, maize cultivation, frequency of antenatal care visit, age of breastfeeding child [8], dietary diversity [9], family size, age at first pregnancy, home delivery, nutrition education [11] mothers' level of education, sickness and production of staple crops [16] were associated with nutritional status of lactating women. However, these factors may vary from setting to setting. Besides, studies which assessed both dietary diversity and nutritional status of lactating mothers in Ethiopia are scant. In this study, we assessed dietary diversity, nutritional status and associated factors of lactating mothers who visited government health facilities at Dessie town, Amhara region, Ethiopia. The output of this study will be important for various stakeholders who are working in improving maternal and child nutrition in the study area.

## Methods

### Study setting and population

Institutional based cross-sectional study was conducted in an urban setting at Dessie town from March to April 2017. Dessie is located about 401 km away from Addis Ababa, the capital city of Ethiopia and 480 kms away from the capital city of the Amhara Regional State, Bahir-Dar. Dessie is one of the three metropolitan towns in the Amhara region. According to Dessie town administration office report, in 2011, Dessie town had a total population of 154,513 of

which 80,575 were females and 73,938 were males. The town has 5 governmental health facilities; 1 referral hospital and 4 health centers. Our study participants were lactating mothers/ breastfeeding mothers (15–49 years) with children under two years who visited Dessie town health facilities during the study period. Lactating mothers visited these health facilities to get various services such as family planning services and vaccination services for their children. Lactating mothers who were critically ill, had physical deformity (that causes difficulty for anthropometric measurements), and who were pregnant during the study period were excluded.

## Sample size and sampling procedures

Sample size was determined using single population proportion formula by considering the following assumptions: 56.4% proportion of lactating mothers with inadequate diet diversity score [14], 95% confidence interval and 5% margin of error. A sample size of 416 was taken after considering 10% non response rate. Systematic random sampling technique was employed to select mothers after the first eligible lactating woman was selected by lottery method. In this regard, every 2$^{nd}$ (K = 1.7) lactating woman visiting the health facilities was included in the study. This was determined by calculating the average monthly flow of lactating mothers for three months to each health facility (*i.e.* 163+175+214+50+110 = 712/ 416 = 1.71).

## Data collection

Data were collected using pre-tested and interviewer-administered questionnaire adapted from different literatures. The questionnaire was used to collect socio-demographic and economic characteristics, health related characteristics, food security status and dietary diversity of participants. It was first prepared in English, translated into Amharic and translated back to English by another person to check its consistency. The translated Amharic version was pre-tested on 21 (5%) of similar subjects at Dessie Town Family Guidance Association model clinic to ensure appropriateness of the study tools and to acquire common understanding on the assessment tools. During data collection, four nurses were hired as data collectors and 2 health officers were involved as supervisors. Data collectors and supervisors were trained for two days on the study objectives, purpose and how to take anthropometric measurements based on the research instrument.

Food insecurity was assessed using household food insecurity access scale (HFIAS) version 3 [17], a tool validated in Ethiopia [18] as well as other developing countries [19, 20]. The HFIAS tool has nine questions asking household's last month experience about three domains of food insecurity: feeling uncertainty of food supply, insufficient quality of food, and insufficient food intake and its physical consequences. Study participants were categorized into two levels of food-security status (food-secured and food-insecured) [21] as follows; they were classified as food secure if the participants responded 'no' to all of the nine questions and insecure if the participants responded 'yes' to at least one of the 9 questions included on the HFIAS tool.

Dietary diversity of lactating mothers was assessed using a 24-hour dietary recall method. Participants were asked to recall freely what they consumed the previous day, inside and outside their home. We then categorized the foods they consumed into the nine food groups (starchy staples, roots and tubers; dark green leafy vegetables; other vitamin A rich fruits and vegetables; other fruits and vegetables; fats and oils; meat and fish; eggs; legumes; nuts and seeds and milk and milk products) [22]. Dietary diversity score (DDS) was determined as the sum of the number of different food groups consumed by the mother in the 24 hours prior to the assessment. Mothers were categorized as having adequate or inadequate dietary diversity

after calculating the mean DDS. Mothers who had consumed food groups below the mean DDS were considered as having inadequate DDS and those who consumed higher or equal to the mean DDS were considered as having adequate DDS. In our case, mothers who consumed < 5.3 mean food groups were considered as having inadequate dietary diversity and those who consumed ≥5.3 mean food groups were considered as having adequate dietary diversity.

Anthropometric measurement (weight and height) of lactating mothers was taken using a weighing scale with an attached height meter (Charder HM200P Stadiometer, Taiwan). During anthropometric measurements, mothers removed their shoes and wore light clothing. The weighing scale was checked before and after each measurement for its accuracy by an object with a known weight. Body mass index (BMI) was then calculated by dividing the weight of mothers in kilogram to height in meter square ($kg/m^2$). BMI was calculated using CDC's online BMI calculator for adults and was also checked manually. For mothers with age below 18years, BMI for age was calculated.

## Data analysis

Data were cleaned, coded and entered into EPI-INFO version 3.5.4 software and transferred and analyzed using SPSS version 22. Descriptive statistics such as frequencies, proportions and chi-square ($X^2$) were used to present the study results. In this study, there were two dependent variables; dietary diversity and nutritional status of lactating mothers. In the binary logistic regression analysis, the association between single explanatory variables and dependent variable was examined by computing odds ratio at 95% confidence level. Independent variables with p-value less than 0.2 were fitted in to a multivariate logistic regression model to identify factors associated with dependent variables. For all statistical significance tests between each independent and dependent variables, significance level was declared if p-value was < 0.05.

## Ethics approval and consent to participate

The study protocol was approved by the Ethical Review Board of Faculty of Chemical and Food Engineering, Bahir Dar University. Permission to conduct the research was granted by Amhara Region Health Bureau, Dessie Referral Hospital and Dessie town health department. Informed consent was obtained from participants after explaining the study objectives. Participation was voluntary and mothers signed (or provided a thumb print if illiterate) a statement of an informed consent after which they were interviewed. For participants who were below 18 years old, written consent was secured from them and from their guardian as well.

## Results

### Socio-demographic characteristics

A total of 408 lactating mothers participated in this study making a response rate of 98.1%. The few non-response rates were due to refusal to participate in the study. The mean (± SD) age of lactating mothers was 26.1 (±4.5) years. About 81% of participants attended formal education and more than half (59.3%) of them had a monthly household income of more than 2000 Ethiopian Birr. The majority (79.4%) of study participants were housewives; married (98.5%) and live in male-headed households (64%) (Table 1).

### Eating habits, dietary diversity and food security

Table 2 presents eating habits, dietary diversity and food security status of lactating mothers. Lactating mothers were asked if there were any changes in their eating habits such as changes

**Table 1. Socio-demographic and economic characteristics of lactating mothers (n = 408) visiting governmental health facilities of Dessie town, Ethiopia, March-April, 2017.**

| Characteristics | Number | Percent |
|---|---|---|
| **Age groups (in years)** | | |
| 15–19 | 17 | 4.2 |
| 20–29 | 303 | 74.2 |
| 30–40 | 88 | 21.6 |
| Mean (±SD) maternal age in years | 26.1 (± 4.5) | |
| **Maternal religion** | | |
| Muslim | 241 | 59.1 |
| Orthodox | 164 | 40.2 |
| Protestant | 3 | 0.7 |
| **Residence** | | |
| Urban | 392 | 96.1 |
| Rural | 16 | 3.9 |
| **Maternal education** | | |
| No formal Education | 76 | 18.6 |
| Primary Education (Grade 1–8) | 131 | 32.1 |
| Secondary Education (Grade 9–12) | 127 | 31.1 |
| College Diploma & above | 74 | 18.1 |
| **Husband education** | | |
| No formal Education | 46 | 11.3 |
| Primary Education (Grade 1–8) | 98 | 24.0 |
| Secondary Education (Grade 9–12) | 123 | 30.1 |
| College Diploma & above | 141 | 34.6 |
| **Maternal occupation** | | |
| House wife | 326 | 79.9 |
| Daily laborer | 8 | 2.0 |
| Merchant | 23 | 5.6 |
| Private Business | 18 | 4.4 |
| Government Employee | 31 | 7.6 |
| NGO Employee | 2 | 0.5 |
| **Husband occupation** | | |
| No work | 7 | 1.7 |
| Daily laborer | 37 | 9.1 |
| Merchant | 103 | 25.2 |
| Private Business | 129 | 31.6 |
| Government Employee | 123 | 30.1 |
| NGO Employee | 9 | 2.2 |
| **Household monthly income (in ETB)** | | |
| ≤ 500 | 7 | 1.7 |
| 501–1000 | 64 | 15.7 |
| 1001–1500 | 42 | 10.3 |
| 1501–2000 | 53 | 13.0 |
| > 2000 | 242 | 59.3 |
| **Type of house** | | |
| Corrugated iron roof wall made with soil | 287 | 70.3 |
| Corrugated iron roof wall made with cement | 121 | 29.7 |
| **Head of household** | | |

(*Continued*)

**Table 1.** (Continued)

| Characteristics | Number | Percent |
|---|---|---|
| Husband | 261 | 64.0 |
| Wife | 17 | 4.2 |
| Both Husband & wife | 130 | 31.9 |
| **Current marital status** | | |
| Married/Living together | 402 | 98.5 |
| Single/ Never married | 1 | 0.2 |
| Divorced/separated/Widowed | 5 | 1.2 |
| **Family size** | | |
| 1–3 persons | 207 | 50.7 |
| 4–6 persons | 187 | 45.8 |
| > 6 persons | 14 | 3.4 |

in meal frequency; in their food intake and avoidance of any kind of foods during their lactation period. In this regard, only 46.3% of lactating women consumed 4 or more times per day and the majority (65.7%) didn't change their food intake during lactation.

The mean (±SD) dietary diversity score of lactating mothers was 5.3 (±1.74) and more than half (55.6%) of them had inadequate dietary diversity (DDS less than 5.3). Food groups such as fats and oils (98.3%) and starchy staples, roots and tubers (89.2%) were the most consumed food groups by the mothers. About three fourth of the mothers had consumed legumes, nuts and seeds and 65% of mothers had consumed dark green leafy vegetables. Compared to other food groups, animal source foods such as meat, fish, eggs and milk were the least consumed food groups (consumed by less than 40% of the mothers). More than one fourth (29.2%) of lactating mothers participated in our study were food insecured.

Lactating mothers were also asked if they have got any information related to nutrition (such as feeding during pregnancy and lactation, consumption of diversified food items, inclusion of fruit and vegetables in the diet, micronutrient supplementation etc). In this regard, the majority (64.5%) of lactating mothers had nutrition information and half of these mothers have got this information from health professionals during their antenatal care visits (Table 2).

## Nutritional status of lactating mothers

The mean BMI (±SD) of lactating mothers was 22.5(±3.5) kg/m$^2$. About 21% of mothers were undernourished (BMI less than 18.5 kg/m$^2$) and 3.68% mothers were obese (Fig 1).

## Factors associated with dietary diversity of lactating mothers

In the bivariate analysis, maternal educational status, husband education, household monthly income, type of house, daily meal frequency, changes in food intake during lactation, nutrition information, and food security status of lactating mothers had association with dietary diversity (Table 3). However, in the multivariable logistic regression analysis, household monthly income, type of house, nutrition information, and food security status of lactating mothers were factors which showed association with dietary diversity of lactating mothers. Lactating mothers who had household monthly income of less than or equal to 1,500 ETB were 2 times more likely to have low dietary diversity than those who had household monthly income of more than 1,500 ETB [AOR = 2.0, 95% CI (1.15, 3.65)]. Similarly, lactating mothers who lived in corrugated iron roof and wall made of soil were 1.8 times more likely to have low dietary diversity than those who lived in a house with corrugated iron roof and wall made of cement

**Table 2. Eating habits, dietary diversity and food security status of lactating mothers (n = 408) visiting governmental health facilities, Dessie town, Ethiopia, March-April, 2017.**

| Characteristics | | Number | Percent |
|---|---|---|---|
| Daily Meal Frequency | | | |
| 2 times | | 24 | 5.9 |
| 3 times | | 195 | 47.8 |
| 4 & above times | | 189 | 46.3 |
| Changes in food intake during lactation | | | |
| Yes | | 140 | 34.3 |
| No | | 268 | 65.7 |
| Food intake changes | | | |
| Frequency of meal | | 36 | 8.8 |
| Amount of meal | | 49 | 12.0 |
| Both frequency & amount of meal | | 54 | 13.2 |
| Avoidance of food during lactation | | | |
| Yes | | 31 | 7.6 |
| No | | 377 | 92.4 |
| Got nutrition information | | | |
| Yes | | 262 | 64.2 |
| No | | 146 | 35.8 |
| Source of nutrition information | | | |
| Health professionals | | 204 | 50.0 |
| Mass media | | 44 | 10.8 |
| Both health professionals and mass media | | 14 | 3.4 |
| Food groups consumed by lactating mothers in previous 24 hours | | | |
| Starchy staples, roots and tubers | | 364 | 89.2 |
| Dark green leafy vegetables | | 265 | 65.0 |
| Other vitamin A rich fruits and vegetables | | 170 | 41.7 |
| Other fruits and vegetables | | 209 | 51.2 |
| Fats and oils | | 401 | 98.3 |
| Meat and fish | | 146 | 35.8 |
| Eggs | | 149 | 36.5 |
| Legumes, nuts and seeds | | 309 | 75.7 |
| Milk and milk products | | 159 | 39.0 |
| Mean dietary diversity score | | 5.3±1.74 | |
| Food security status | Food secured | 289 | 70.8 |
| | Food insecured | 119 | 29.2 |

[AOR = 1.8, 95% CI (1.15, 2.94)]. Nutrition information had also a significant association with dietary diversity of lactating mothers. Lactating mothers who did not get nutrition information were 1.6 times more likely to have low dietary diversity compared to those who have got nutrition information [AOR = 1.6, 95% CI (1.05, 2.61)]. Lactating mothers who lived in food insecured households were 1.8 times more likely to have low dietary diversity than those who lived in food secured households [AOR = 1.8, 95% CI (1.05, 3.06)] (Table 3).

## Factors associated with nutritional status of lactating mothers

Both bivariate and multivariate analyses were done to identify factors associated with nutritional status of lactating mothers (Table 4). In the bivariate analysis, maternal age, marital

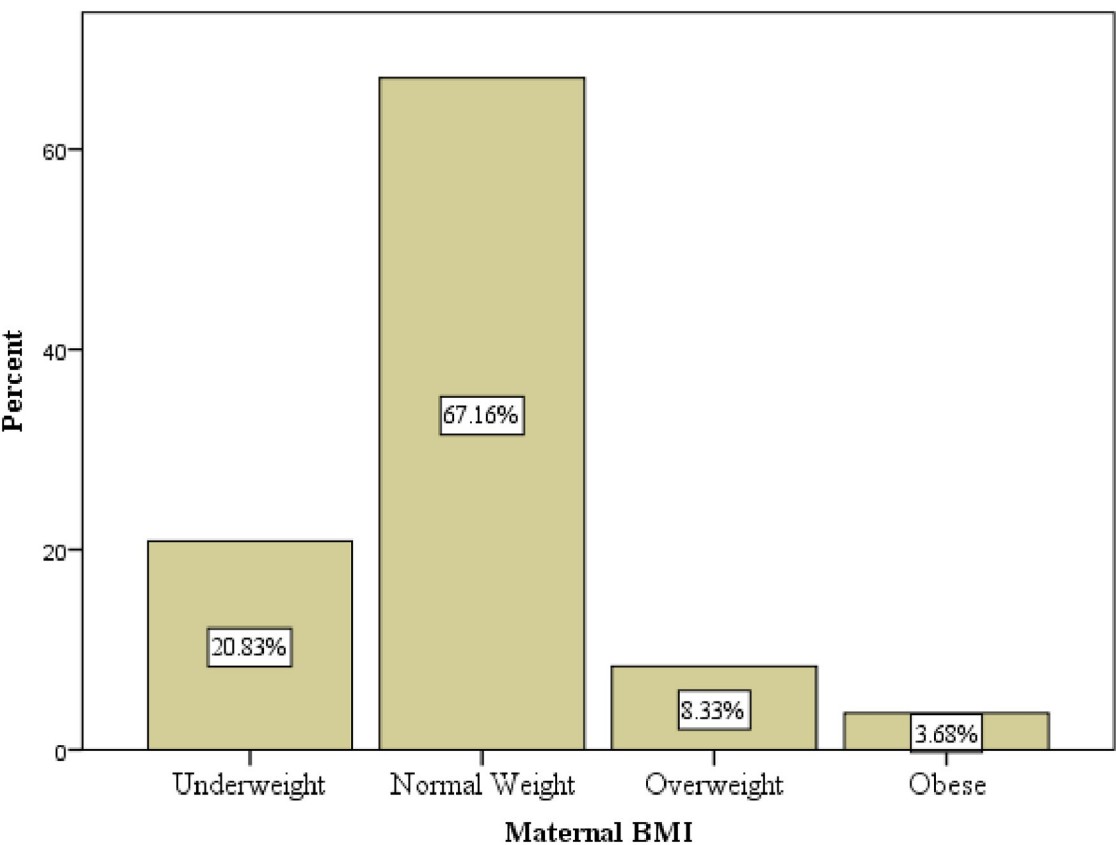

**Fig 1.**

status, husband education, household monthly income, daily meal frequency, nutrition information, household food security status and women dietary diversity had association with nutritional status of lactating mothers.

In the multivariable logistic regression analysis, maternal age, marital status, inadequate dietary diversity and household food insecurity showed association with nutritional status of lactating mothers. Lactating women in the age group of 15–19 years and 20–29 years were 10.3 times [AOR = 10.3, 95% CI (2.89, 36.39)] and 3.4 times [AOR = 3.4, 95% CI (1.57, 7.36)] more likely to be underweight than older mothers respectively. Maternal marital status was also one of the factors which showed association with mothers' nutritional status. Lactating women who were divorced or separated were 10 times more likely to be undernourished than their counterparts [AOR = 10.1, 95% CI (1.42, 72.06)].

Lactating mothers who had inadequate dietary diversity score were 3.8 times more likely to be undernourished than those who had adequate dietary diversity score [AOR = 3.8, 95% CI (2.08, 7.03)]. Similarly, lactating mothers who lived in food insecured households were 3 times at risk of becoming undernourished compared to their counterparts [AOR = 3.1, 95% CI (1.81, 5.32)].

## Discussion

The mean dietary diversity score (DDS) of lactating mothers in our study was 5.3 and this was slightly higher than studies reported from other parts of Ethiopia; Jimma zone (4.9) [9] and

**Table 3. Association of variables with dietary diversity of lactating mothers (n = 408) visiting governmental health facilities of Dessie town, Ethiopia, March-April, 2017.**

| Variables | Dietary Diversity | | COR | AOR |
|---|---|---|---|---|
| | Inadequate | Adequate | | |
| | n (%) | n (%) | (95% CI) | (95% CI) |
| **Maternal age (in years)** | | | | |
| 15–19 | 12(2.96) | 5 (1.24) | 1 | 1 |
| 20–29 | 156(38.2) | 147(36.0) | 0.5(0.32, 0.86) | 0.5(0.15, 1.50) |
| 30–40 | 59(14.48) | 29(7.12) | 1.2(0.38, 3.67) | 0.8(0.25, 2.84) |
| **Current marital status** | | | | |
| Married/Living together | 223(54.64) | 179(43.86) | 1 | 1 |
| Divorced/Separated/ | 4(0.92) | 2(0.48) | 1.65(0.29, 8.87) | 0.4(0.05, 3.33) |
| **Maternal education** | | | | |
| No formal Education | 54 (13.2%) | 22 (5.4%) | 2.3 (1.31, 3.87)** | 1.5 (0.82, 2.83) |
| Formal Education | 173 (42.4%) | 159 (39.0%) | 1 | 1 |
| **Husband education** | | | | |
| No formal Education | 34 (8.3%) | 12 (3.0%) | 2.5 (1.25, 4.95)** | 0.8 (0.30, 1.85) |
| Formal Education | 193 (47.3%) | 169 (41.4%) | 1 | 1 |
| **Household monthly income** | | | | |
| ≤ 1500 ETB | 85 (20.8%) | 28 (6.9%) | 3.3 (2.02, 5.31)** | 2.0 (1.15, 3.65)** |
| > 1500 ETB | 142 (34.8%) | 153 (37.5%) | 1 | 1 |
| **Type of house** | | | | |
| Corrugated iron roof and wall made with soil | 177 (43.4%) | 110 (27.0%) | 2.3 (1.48, 3.52)** | 1.8 (1.15, 2.94)** |
| Corrugated iron roof and wall made with cement | 50 (12.2%) | 71 (17.4%) | 1 | 1 |
| **Nutrition information** | | | | |
| Yes | 132 (32.3%) | 130 (31.9%) | 1 | 1 |
| No | 95 (23.3%) | 51 (12.5%) | 1.8 (1.21, 2.79)** | 1.6 (1.05, 2.61)** |
| **Daily meal frequency** | | | | |
| ≤ 3 Meals/day | 136 (33.3%) | 83 (20.4%) | 1.8 (1.19, 2.62)** | 1.3 (0.86, 2.02) |
| > 3 Meals/day | 91 (22.3%) | 98 (24.0%) | 1 | 1 |
| **Changes in food intake** | | | | |
| Yes | 67 (16.4%) | 73 (17.9%) | 1 | 1 |
| No | 160 (39.2%) | 108 (26.5%) | 1.6 (1.07, 2.44)** | 1.3 (0.80, 1.95) |
| **Food security status** | | | | |
| Food Secured | 142 (34.8%) | 147 (36.1%) | 1 | 1 |
| Food Insecured | 85 (20.8%) | 34 (8.3%) | 2.6 (1.63, 4.10)** | 1.8 (1.05, 3.06)** |

COR- Crude Odds Ratio, AOR- Adjusted Odds Ratio, ETB-Ethiopian birr N.B- *p- value significant at level of P <0.2,

**p-value significant at level of P<0.05.

Aksum town (3.4) [14]. These differences might be due to differences in socio-demographic and economic situations of mothers.

The majority (98%) of lactating mothers in our study reported that they have consumed oils and fats in the previous 24 hours and this is related to the tradition of adding small amount of oil or fat (commonly butter) in the preparation of Ethiopian stews or dishes at least three times a day. Starchy staples, roots and tubers were also the most consumed food groups (nearly by 90% of the mothers) and this is in agreement with other studies reported from different parts of Ethiopia [8, 13, and 14].

**Table 4. Association of variables with nutritional status of lactating mothers (n = 408) visiting governmental health facilities of Dessie town, Ethiopia, March-April, 2017.**

| Variables | Nutritional status (BMI) | | COR | AOR |
| --- | --- | --- | --- | --- |
| | Underweight | Normal/Overweight/Obese | | |
| | n (%) | n (%) | (95% CI) | (95% CI) |
| **Maternal age (in years)** | | | | |
| 15–19 | 8(2.0) | 9(2.2) | 6.2(1.98, 19.51) | 10.3(2.89, 36.39)** |
| 20–29 | 66(16.2) | 237(58.0) | 1.9(0.98, 3.88) | 3.4(1.59, 7.36)** |
| 30–40 | 11(2.7) | 77(18.9) | 1 | 1 |
| **Maternal marital Status** | | | | |
| Married/Living together | 81(19.8) | 321(78.7) | 1 | 1 |
| Divorced/separated | 4(0.9) | 2(0.5) | 7.9(1.43, 44.03) | 10.1 (1.42, 72.06)** |
| **Head of household** | | | | |
| Husband | 61 (15.0) | 200(49.0) | 1.9(1.07, 3.37) | 1.4(0.71, 2.66) |
| Wife | 6(1.5) | 11(2.7) | 3.4(1.15, 10.32) | 0.8(0.18, 3.79) |
| Both husband & wife | 18(4.4) | 112(27.5) | 1 | 1 |
| **Maternal education** | | | | |
| No formal education | 19(4.7) | 57(13.9) | 1.3(0.75, 2.41) | 0.5(0.24, 1.19) |
| Formal education | 66(16.2) | 266(65.2) | 1 | 1 |
| **Husband education** | | | | |
| No formal education | 19 (4.6) | 27 (6.6) | 3.2 (1.66, 6.01)** | 1.9(0.88, 3.98) |
| Formal education | 66 (16.2) | 296 (72.6) | 1 | 1 |
| **Household monthly income** | | | | |
| ≤ 1500 ETB | 38 (9.3) | 75 (18.4) | 2.7(1.62, 4.41)** | 1.1 (0.54, 2.34) |
| > 1500 ETB | 47 (11.5) | 248 (60.8) | 1 | 1 |
| **Type of house** | | | | |
| Corrugated iron roof wall made with soil | 67(16.4) | 220(53.9) | 1.7(0.96, 3.08) | 1.4(0.73, 2.73) |
| Corrugated iron roof wall made with cement | 18(4.4) | 103(25.3) | 1 | 1 |
| **Daily meal frequency** | | | | |
| ≤ 3 meals/day | 56 (13.7) | 163 (40.0) | 1.9 (1.15, 3.12)** | 1.5 (0.84, 2.58) |
| > 3 meals/day | 29 (7.1) | 160 (39.2) | 1 | 1 |
| **Avoidance of food during lactation** | | | | |
| Yes | 5(1.2) | 26(6.4) | 0.7 (0.27, 1.92) | 1.1 (0.32, 3.22) |
| No | 80(19.6) | 297(72.8) | 1 | 1 |
| **Family size** | | | | |
| 1–3 persons | 50(12.2) | 157(38.5) | 1 | 1 |
| 4–6 persons | 32(7.8) | 155(38) | 0.6(0.39, 1.07) | 1.2 (0.52, 2.91) |
| > 6 persons | 3(0.7) | 11(2.7) | 0.9(0.23, 3.19) | 2.9 (0.52, 16.90) |
| **Nutrition information** | | | | |
| Yes | 47 (11.5) | 215 (52.7) | 1 | 1 |
| No | 38 (9.3) | 108 (26.5) | 1.6 (0.99, 2.62)* | 1.2 (0.66, 2.13) |
| **Women dietary diversity** | | | | |
| Adequate | 17 (4.2) | 164 (40.2) | 1 | 1 |
| Inadequate | 68 (16.6) | 159 (39.0) | 4.1 (2.32, 7.33)** | 3.8 (2.08, 7.03)** |
| **Food security status** | | | | |
| Food secured | 41 (10.0) | 248 (60.8) | 1 | 1 |
| Food insecured | 44 (10.8) | 75 (18.4) | 3.5 (2.16, 5.84)** | 3.1(1.81, 5.32)** |

COR- Crude Odds Ratio, AOR- Adjusted Odds Ratio, ETB-Ethiopian birr,

**p-value significant at level of P<0.05.

The mean BMI of lactating mothers was 22.5 kg/m$^2$. This figure was slightly higher than the mean BMI of lactating women reported from Womberma woreda of Amhara region (20 kg/m$^2$) [11] and Jimma zone, Oromia region, Ethiopia (19.2 kg/m$^2$) [9]. These differences might be due to differences in socio demographic and economic characteristics of study participants.

Nearly one fifth of our study participants (20.8%) were undernourished (BMI less than 18.5kg/m$^2$). This prevalence was comparable with that reported for lactating women who attended Nekemtie town hospitals and health centers (20.5%) [10]. On the other hand, the prevalence of undernutrition in our study was lower than that reported from Samre woreda (31%) [8] and Alamata district of Tigray, Ethiopia (24.6%) [12]. It is recommended that lactating woman should take at least two additional meals per day during lactation [23]. However, in our study more than half of the mothers didn't take any additional meal during lactation which may result in low dietary intakes. Dietary intakes below the recommended frequency might lead mothers to poor nutritional status. In general, poor nutritional status of lactating women is a developmental threat of a given country as children born from women who became malnourished during pregnancy and lactation are at higher risk of developing various health problems [24].

Lactating mothers who had household monthly income of less than or equal to 1,500 ETB were two times more likely to have low dietary diversity than those who had household monthly income of more than 1,500 ETB. This finding is in agreement with a study conducted in Aksum town, Ethiopia [14] and a study conducted in Bangladesh [25]. This might be due to the fact that having low monthly income hinders lactating mothers from purchasing diversified foods. Similarly, lactating mothers who lived in corrugated iron roof with wall made of soil were 1.8 times more likely to have low dietary diversity than those who lived in a house with corrugated iron roof wall made of cement. This might be associated with the economic status of the households' as living in an improved house can be directly related to the economic status of lactating mothers and high probability of having a diversified food.

In our study, mothers who did not get nutrition information were 1.6 times more likely to have low dietary diversity than those mothers who got nutrition information. Unlike other studies which showed a positive association between education and dietary diversity [13, 26], education by itself didn't have association with mothers' dietary diversity in our study. This finding shows that rather than formal education, specific information about nutrition is the one which helps mothers to improve their dietary pattern or eat a diversified diet. In fact, Woldehawaria et al. [14] from Aksum town, Ethiopia also indicated absence of association between education and maternal dietary diversity.

Lactating mothers who lived in food insecured households were 1.8 times more likely to have low dietary diversity than those who lived in food secured households. A study done in Angecha district, Southern Ethiopia also reported that mothers from food-insecure households were 3.4 times more likely to have low dietary diversity [26] when compared with mothers from food secure households. Reports from other countries such as Vietnam, Bangladesh and Nepal [13, 25, 27] also support our finding. On the other hand, food insecurity had no association with dietary diversity in a study conducted in Aksum town, Ethiopia [14].

The covariates maternal age, marital status, women dietary diversity and household food security status had statistically significant association with mothers' nutritional status. Young mothers and mothers who were divorced or separated had a higher chance of being undernourished than their counterparts. Similar finding was reported by Teller and Yimer [28] from Southern Ethiopia. This might be associated with the economic status of mothers as it could be endangered by a negative change in marital status. Lactating mothers with inadequate dietary diversity were 3.8 times more likely to be exposed to undernutrition compared to those who had adequate dietary diversity. This was supported by a study conducted in Dedo and Seqa-Chekorsa

Districts of Jimma Zone, Ethiopia [9]. Similarly, lactating mothers from food insecure households were 3 times more likely to be undernourished when compared with those mothers from food secure households. Our finding was supported by one study from rural Kenya [29]. Different studies also indicated the association between household food insecurity with inadequate energy and nutrient intake and in turn malnutrition among household members [19, 30].

Our study had two major limitations due to its cross sectional nature; one it was not possible to assess seasonal variation of food availability which will have an effect on dietary diversity and two it was difficult to establish a cause and effect relationship between one of our dependent variables (nutritional status) and the independent variables although some associations were observed.

## Conclusion

The dietary diversity of lactating mothers in the study area was sub optimal and prevalence of undernutrition was high. Household monthly income, type of house, nutrition information, and household food insecurity status were factors significantly associated with dietary diversity of lactating mothers. On the other hand, inadequate dietary diversity and food insecurity were factors strongly associated with the nutritional status of lactating mothers. Public health nutrition interventions such as improving accessibility of affordable and diversified nutrient rich foods are important to improve the nutritional status of mothers and their children in the study area.

## Supporting information

**S1 Data. Manuscript data.**
(DOC)

**S2 Data. Supplementary data.**
(SAV)

**S1 Questionnaire.**
(DOC)

## Acknowledgments

We are grateful for Bahir Dar Institute of Technology, School of Research and Graduate Studies for supporting this study. We are also grateful to Amhara Region Health Bureau and Dessie town health department for facilitating this research by timely writing support letters. Finally, our special thanks go to data collectors and study participants who contributed to this study.

## Author Contributions

**Conceptualization:** Awel Seid, Hirut Assaye Cherie.

**Data curation:** Awel Seid.

**Formal analysis:** Awel Seid, Hirut Assaye Cherie.

**Investigation:** Awel Seid.

**Methodology:** Awel Seid, Hirut Assaye Cherie.

**Project administration:** Awel Seid, Hirut Assaye Cherie.

**Supervision:** Hirut Assaye Cherie.

**Writing – original draft:** Awel Seid, Hirut Assaye Cherie.

**Writing – review & editing:** Awel Seid, Hirut Assaye Cherie.

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
