## [Decision Letter · Decision Letter 0]

9 Jun 2021

PONE-D-20-31562

Dietary diversity, nutritional status and associated factors among lactating mothers visiting government health facilities at Dessie town, Amhara region, Ethiopia

PLOS ONE

Dear Dr. Cherie,

Thank you for submitting your manuscript to PLOS ONE. After careful consideration, we feel that it has merit but does not fully meet PLOS ONE’s publication criteria as it currently stands. Therefore, we invite you to submit a revised version of the manuscript that addresses the points raised during the review process.

I would like to sincerely apologise for the delay you have incurred with your submission. It has been exceptionally difficult to secure reviewers to evaluate your study. We have now received two completed reviews; their comments are available below. The reviewers have raised several concerns about the study that need to be addressed in a revision.

Please revise the manuscript to address all the reviewer's comments in a point-by-point response in order to ensure it is meeting the journal's publication criteria. Please note that the revised manuscript will need to undergo further review, we thus cannot at this point anticipate the outcome of the evaluation process.

We look forward to receiving your revised manuscript.

Kind regards,

Miquel Vall-llosera Camps

Senior Editor

PLOS ONE

Journal Requirements:

1. Please ensure that your manuscript meets PLOS ONE's style requirements, including those for file naming. The PLOS ONE style templates can be found at andhttps://journals.plos.org/plosone/s/file?id=ba62/PLOSOne_formatting_sample_title_authors_affiliations.pdf

3.Thank you for stating the following financial disclosure: School of Research and Graduate Studies, Bahir Dar Institute of Technology funded this research through its program of funding researches conducted by its staff.

a. Please clarify the sources of funding (financial or material support) for your study. List the grants or organizations that supported your study, including funding received from your institution.

b. State what role the funders took in the study. If the funders had no role in your study, please state: “The funders had no role in study design, data collection and analysis, decision to publish, or preparation of the manuscript.

d. If you did not receive any funding for this study, please state: The authors received no specific funding for this work.

5. Thank you for stating the following in your Competing Interests section: "No".

6. We note that you have indicated that data from this study are available upon request. PLOS only allows data to be available upon request if there are legal or ethical restrictions on sharing data publicly. For more information on unacceptable data access restrictions, please see http://journals.plos.org/plosone/s/data-availability#loc-unacceptable-data-access-restrictions.

a. If there are ethical or legal restrictions on sharing a de-identified data set, please explain them in detail (e.g., data contain potentially sensitive information, data are owned by a third-party organization, etc.) and who has imposed them (e.g., an ethics committee). Please also provide contact information for a data access committee, ethics committee, or other institutional body to which data requests may be sent.

b. If there are no restrictions, please upload the minimal anonymized data set necessary to replicate your study findings as either Supporting Information files or to a stable, public repository and provide us with the relevant URLs, DOIs, or accession numbers. For a list of acceptable repositories, please see http://journals.plos.org/plosone/s/data-availability#loc-recommended-repositories.

Reviewers' comments:

Reviewer's Responses to Questions

**Comments to the Author**

1. Is the manuscript technically sound, and do the data support the conclusions?

Reviewer #1: Partly

Reviewer #2: No

2. Has the statistical analysis been performed appropriately and rigorously? 

Reviewer #1: No

Reviewer #2: No

3. Have the authors made all data underlying the findings in their manuscript fully available?

Reviewer #1: No

Reviewer #2: No

4. Is the manuscript presented in an intelligible fashion and written in standard English?

Reviewer #1: No

Reviewer #2: Yes

5. Review Comments to the Author

Reviewer #1: Comments:

1. It is important that this study highlights the important issues faced by lactating mothers, especially in resource poor context. The manuscript will therefore be of interest to those who are working in the related fields, and I hope can be published in the PLOS ONE after a revision.

2. Background: The authors have attempted to highlight the importance of food security and dietary diversity. However, I could not find a clear rationale why the authors wanted to focus on both the outcomes at the same time in this manuscript. In addition, it would be helpful if the authors state the scientific gaps and how this study is going to fill the gaps. Why do the authors think that this study is beneficial for researchers and the public in other regions within and outside the country? Further, the authors have mentioned “lactating mothers” often, but did not explain what they mean by “lactating mothers” in this study.

3. Line 56-57- reference is missing?

4. Line 67-69, when these study were conducted?

5. Provide theoretical basis for selecting independent variables in the study? What theories underpin your study?

6. The HFIAS tools was validated in Tanzania and Iran, how would you justify the use of this tool in Ethiopia?

7. What are the possible bias in the study and how it was attempted to minimize it?

8. Why study participants were categorized only into two levels of food-security 123 status (food-secured and food-insecure)? Why not the cumulative HFIAS score was categorized into four levels of household food insecurity: food secured, and mild, moderate, and severe food insecurity, following HFIAS guideline.

9. What does nutrition information refers to?

10. What happen after pretesting of tools?

11. Line 136-38, what is your reference to take this cut of value for low and high DDS.

12. “For mothers with age below 18years, BMI for age was calculated” what reference you used for this measurement?

13. Why all independent variables were fitted in to a multivariate logistic regression model to identify factors associated with dependent variables? Why not only significant variables?

14. In table 1, need to mention what does” P” “a” “b” stands for , and where does it come from?

15. Also, the p-value 0.00 need to be presented in standard form for writing p-value.

16. P-value in all tables need to be presented in standard format.

17. Digit after decimal need to be uniform though out the manuscript.

18. Discussion need to focus on major findings of the study and also please revise this section thoroughly and provide sufficient discussion of relevant studies.

19. Please also check this study: “Food insecurity and dietary diversity among lactating mothers in the urban municipality in the mountains of Nepal”. https://journals.plos.org/plosone/article/authors?id=10.1371/journal.pone.0227873

20. Language correction is required. I suggest author to have proof read of the manuscript from native English language speaker.

21. Could you please also provide your data set for review purpose?

Reviewer #2: The effort in this paper is good at providing an overview of the problems with diet and nutritional status during lactation. This information is also essential due to the lack of such data in developing countries. However, several things need to be clarified in this manuscript, including:

Characteristics of participant:

The author can explain the activities carried out by lactating women when visiting health facilities, whether the health facility provides education and counselling services during breastfeeding (so we can assume participants are healthy people) or treated for an illness. However, if the participant is sick, the conclusions in the text need to be explained more specifically that the importance of this paper is (for example) to improve the quality of services and education in health facilities, not to the public.

Research purposes: In line 77-79: “However, these factors may not be consistent in all settings and thus call for the need for context-specific information to design and implement appropriate nutrition interventions”, this statement looks inconsistent and hard to follow. The author can explain the urgency of this paper when compared with other existing data. If it is said that the factors related to dietary diversity and nutritional status are not consistent across all settings, the authors are expected to explain why this study was carried out in a more specific context, not in general.

Dietary diversity:

• Authors need to review how to interpret DDS. For example, is DDS data normally distributed or is it necessary to use distribution data.

• Further insights for analyzing DDS can be found in the FAO Guideline (on REF #24, page 26-27). There are no established cut-off points in terms of the number of food groups to indicate adequate or inadequate (or low/high in this text) dietary diversity for the DDS. The author can analyze using the score data from each participant to see the correlation with other variables.

• If the mean DDS used as the cut-off, this would result in a low/high proportion of around 50 per cent. However, the authors need to reconsider the results and discussions regarding the prevalence of low DDS since it cannot be compared with other populations (in line 268-273).

• In line 260-266 Discussion, the authors compare DDS in studies with different maximum DDS values. Please compare something equivalent.

• In line 278-282: oil and fat consumption were high (98%), it mentions due to adding a small amount of oil/fat in meal preparation. The author needs to explain whether there is a restriction of food quantities to at least 15 grams to include the food group in daily consumption. For women aged 15-49 years, dietary diversity scores were more strongly correlated with micronutrient adequacy of the diet when food quantities of approximately one tablespoon or less (<15g) were not included in the score (Arimond et al., 2010).

Nutritional status:

• The author needs to review whether the BMI data is normally distributed to be presented as a mean.

• On line 193-194, it says there are 21% underweight and 12% overweight. This needs to be clarified because, in Table 2 and Table 4, all participants are categorized as underweight and normal.

• The author can also mention whether there are participants who fall into the obese category.

• In addition, similar to DDS, the authors need to consider analyzing the correlation using the continuous variable (BMI itself) compared to the analysis after being categorized.

Dietary assessment method:

• Based on the level of the objective of dietary assessment, the authors need to explain whether the data were collected by single or replicated in non-consecutive days.

• In addition, it is necessary to clarify the method mentioned (24h dietary recall) to record all food consumed by the mother for 24 hours or recall the specific consumption of 9 food groups.

Data analysis:

• It is recommended that the authors present the results of the correlation analysis (r and p values) for each of the tested independent variables, as stated in lines 213-215. (This might be attached in an supplementary table).

• The variables included in the regression analysis also need to be discussed regarding aspects of biological plausibility. For example, if specific variables are tested (such as maternal religion, family size, and head of household), these variables need to be discussed in the introduction/discussion.

Writing suggestions

• In the second paragraph of the Background, the author can select only information related to the topics discussed in this paper. The author needs to reconsider the relationship between urbanization, primary-secondary-tertiary level of health care with the topic.

• Authors can use more recent DHS data (is there a 2016 edition?) to describe nutritional problems in the study area.

• Paternal or parental education?

• Tables 1 & 2, contents and headings in tables are inconsistent. The author needs to review whether the DD & nutritional status data in table 1 is needed. The same data has shown in tables 3 & 4

• Authors need to add information to the superscript “a” and “b” in data tables 1 and 2.

• Eating habits: this data appears in the result, but there is no explanation regarding the meaning of habits.

• Dietary diversity categories: low/high or adequate/inadequate?

• Discussion: In the first paragraph, the author can explain the most interesting findings or the ones that answer the main problem in the research

6. PLOS authors have the option to publish the peer review history of their article (what does this mean?). If published, this will include your full peer review and any attached files.

Reviewer #1: No

Reviewer #2: No

---

## [Author Response · Author response to Decision Letter 0]

19 Sep 2021

Dear Sir/Madam,

We have revised our manuscript based on each reviewer's comments. We have tried to address each of the reviewer's comments and submitted it is as Responses to reviewers' together with our manuscript. Thanks

Sincerely,

---

## [Decision Letter · Decision Letter 1]

19 Jan 2022

PONE-D-20-31562R1

Dietary diversity, nutritional status and associated factors among lactating mothers visiting government health facilities at Dessie town, Amhara region, Ethiopia

PLOS ONE

Dear Dr. Cherie,

Thank you for submitting your manuscript to PLOS ONE. After careful consideration, we feel that it has merit but does not fully meet PLOS ONE’s publication criteria as it currently stands. Therefore, we invite you to submit a revised version of the manuscript that addresses the points raised during the review process.

We look forward to receiving your revised manuscript.

Kind regards,

Mohammad Hossein Ebrahimi

Academic Editor

PLOS ONE

Journal Requirements:

Reviewers' comments:

Reviewer's Responses to Questions

**Comments to the Author**

1. If the authors have adequately addressed your comments raised in a previous round of review and you feel that this manuscript is now acceptable for publication, you may indicate that here to bypass the “Comments to the Author” section, enter your conflict of interest statement in the “Confidential to Editor” section, and submit your "Accept" recommendation.

Reviewer #2: All comments have been addressed

Reviewer #3: All comments have been addressed

Reviewer #4: (No Response)

2. Is the manuscript technically sound, and do the data support the conclusions?

Reviewer #2: Yes

Reviewer #3: Yes

Reviewer #4: Yes

3. Has the statistical analysis been performed appropriately and rigorously? 

Reviewer #2: Yes

Reviewer #3: Yes

Reviewer #4: Yes

4. Have the authors made all data underlying the findings in their manuscript fully available?

Reviewer #2: Yes

Reviewer #3: Yes

Reviewer #4: Yes

5. Is the manuscript presented in an intelligible fashion and written in standard English?

Reviewer #2: Yes

Reviewer #3: Yes

Reviewer #4: Yes

6. Review Comments to the Author

Reviewer #2: The author has made considerable improvements in this paper. Data and supplementary tables are also presented as needed.

Reviewer #3: It is clear that the authors have corrected and improved the manuscript. They also provided responses to all the reviewer's comments. The manuscript is relevant and provides important data to Public Health System of Ethiopia and other developing countries.

Reviewer #4: Based on table 1, one person was single/never married! How was she lactating then?

In table 2, please put SD for mean of dietary diversity score.

For tables 3 and 4, please spelll out AOR and COR at foot of the tables.

7. PLOS authors have the option to publish the peer review history of their article (what does this mean?). If published, this will include your full peer review and any attached files.

Reviewer #2: **Yes: **Sofa Rahmannia

Reviewer #3: No

Reviewer #4: No

---

## [Author Response · Author response to Decision Letter 1]

28 Jan 2022

We have included our responses to reviewer's comments. Thanks

---

## [Editor Report · Decision Letter 2]

2 Feb 2022

Dietary diversity, nutritional status and associated factors among lactating mothers visiting government health facilities at Dessie town, Amhara region, Ethiopia

PONE-D-20-31562R2

Dear Dr. Cherie,

We’re pleased to inform you that your manuscript has been judged scientifically suitable for publication and will be formally accepted for publication once it meets all outstanding technical requirements.

Kind regards,

Mohammad Hossein Ebrahimi

Academic Editor

PLOS ONE
---

## [Editor Report · Acceptance letter]

8 Feb 2022

PONE-D-20-31562R2 

Dietary diversity, nutritional status and associated factors among lactating mothers visiting government health facilities at Dessie town, Amhara region, Ethiopia 

Dear Dr. Cherie:

I'm pleased to inform you that your manuscript has been deemed suitable for publication in PLOS ONE. Congratulations! Your manuscript is now with our production department. 

Kind regards, 

on behalf of

Dr. Mohammad Hossein Ebrahimi 

Academic Editor

PLOS ONE